# Modeling Task Effects on Meaning Representation in the Brain via Zero-Shot MEG Prediction

**Mariya Toneva**[* 1,2], **Otilia Stretcu**[* 1], **Barnabás Póczos**[1], **Leila Wehbe**[1,2], **Tom M. Mitchell**[1,2]

[1] Machine Learning Department, Carnegie Mellon University, Pittsburgh, USA

[2] Neuroscience Institute, Carnegie Mellon University, Pittsburgh, USA

mktoneva@cs.cmu.edu, ostretcu@cs.cmu.edu

## Abstract

How meaning is represented in the brain is still one of the big open questions in neuroscience. Does a word (e.g., *bird*) always have the same representation, or does the task under which the word is processed alter its representation (answering *"can you eat it?"* versus *"can it fly?"*)? The brain activity of subjects who read the same word while performing different semantic tasks has been shown to differ across tasks. However, it is still not understood how the task itself contributes to this difference. In the current work, we study Magnetoencephalography (MEG) brain recordings of participants tasked with answering questions about concrete nouns. We investigate the effect of the task (i.e. the question being asked) on the processing of the concrete noun by predicting the millisecond-resolution MEG recordings as a function of both the semantics of the noun and the task. Using this approach, we test several hypotheses about the task-stimulus interactions by comparing the zero-shot predictions made by these hypotheses for novel tasks and nouns not seen during training. We find that incorporating the task semantics significantly improves the prediction of MEG recordings, across participants. The improvement occurs $475 - 550$ms after the participants first see the word, which corresponds to what is considered to be the ending time of semantic processing for a word. These results suggest that only the end of semantic processing of a word is task-dependent, and pose a challenge for future research to formulate new hypotheses for earlier task effects as a function of the task and stimuli.

## 1 Introduction

One of the central goals of artificial intelligence (AI) is to build intelligent systems that understand the meaning of concepts and use it to perform tasks in the real world. Despite the great strides in learning representations, there are still many problems that could benefit from further improvements in understanding and representing *meaning*, such as symbol grounding, common-sense reasoning, and natural language understanding. While machines are limited in these areas, we do have one system that is capable of representing meaning and performing these tasks well: the human brain. Thus, looking to the brain for insights about how we represent and compose meaning may be beneficial.

Studies of meaning representation in neuroscience have revealed that the brain accesses meaning differently depending on the demands of a task [1, 2, 3, 4, 5]. For instance, the recorded brain activity of a participant that observes the word *"cat"* differs according to whether the participant is asked to answer whether *"cat"* is an animal or a vegetable [6]. The difference is shown to occur between

---

[*]Equal contribution and joint lead authorship.

Code available at https://github.com/otiliastr/brain_task_effect.

$400-600$ms after *"cat"* is presented to the participant, a period when it is believed that the brain processes the semantics of the perceived word [7], suggesting an interaction between the task and stimulus meaning. One hypothesis for the interaction that has received some experimental backing is that, in order to solve the task, the brain uses attention mechanisms to emphasize task-relevant information [8, 9, 10, 11, 12]. However, the computational principle behind this attention mechanism is poorly understood, as it can be due to several neural properties, such as an increased response gain, sharper tuning [13], or a tuning shift [11].

In this work, we propose the first computational model that implements precise hypotheses for the interaction between the semantics of tasks and that of individual concepts in the brain, and tests their ability to explain brain activity. We posit that formulating such a computational model will be a helpful step towards specifying a full account of the task-stimulus interactions. Specifically, we study how tasks interact with the semantics of concepts by building models that predict recorded brain activity of people tasked with answering questions (e.g., *"is it bigger than a microwave?"*) about concrete nouns (e.g., *"bear"*). Importantly, the proposed model is able to generalize to previously unseen tasks and stimuli, allowing us to make zero-shot predictions of brain recordings.

Using this computational framework, we show that models that predict brain recordings as a function of the task semantics significantly outperform ones that do not during time windows ($475-550$ms and $600-650$ms) which largely coincide with the end of semantic processing of a word, typically thought to last until $600$ms [7]. This result suggests that only the end of semantic processing of a word becomes task-dependent and that this effect is related to the meaning of the task. We believe that in addition to this result, neuroscientists will also be interested in the ability to computationally compare different hypotheses for the task-stimuli interactions, and we hope that our general problem formulation will benefit future research attempting to study other forms of interaction not considered in this work. Additionally, our work may be of interest to the AI community. Further understanding task effects on concept meaning in the brain may provide insights into building AI models that learn how to combine representations specific to the task with task-invariant representations of concepts, as a step towards composing meaning that is both goal-oriented and more easily adaptable to new tasks.

Our **main contributions** can be summarized as follows:

- We propose a means of representing the semantics of the question task that shows a significant relationship with the elicited brain response. We believe such an approach could be useful to future studies on question-answering in the brain.

- We provide the first methodology that can predict brain recordings as a function of *both* the observed stimulus and question task. This is important because it will not only encourage neuroscientists to formulate mechanistic computational hypotheses about the effect of a question on the processing of a stimulus, but also enable neuroscientists to test these different hypotheses against each other by evaluating how well they can align with brain recordings. While we have implemented and compared several hypotheses for this effect, and have found some to be better than others, parts of the MEG recordings remain to be explained by future hypotheses. We hope neuroscientists will build on our method to formulate and test such future hypotheses. We make our code publicly available to facilitate this.

- We perform all learning in a zero-shot setting, in which neither the stimulus nor the question used to evaluate the learned models is seen during training (i.e. not just as the specific stimulus-question pair but also in combination with any other question/stimulus). Note that this is not the case in previous work that examines task effects, and we are the first to demonstrate how zero-shot learning can be applied successfully to this question. This is important for scientific discovery because it can test the generalization of the results beyond the experimental stimuli and tasks.

- We show that models that integrate task and stimulus representations have significantly higher prediction performance than models that do not account for the task semantics, and localize the effect of task semantics largely to time-windows in $475-650$ms after the stimulus presentation.

## 2 Related work

Classical neuroimaging experiments that study meaning by contrasting different stimulus conditions often include a task that is related to processing the meaning of the word (such as judging the

Figure 1: Experimental paradigm recreated from Sudre et al. [19]. Subjects shown a question, followed by 60 concrete nouns along with their line drawings in random order.

Figure 2: Feature representations of questions and stimuli obtained from Mechanical Turk.

similarity of two stimuli), however these experiments do not use predictive models that systematically relate the stimulus properties to the brain recordings, and do not explicitly investigate the task effect.

A number of previous studies have used predictive models to examine the relationship between brain recordings and stimulus properties, but have also not explicitly investigated the effect of a task. In many of these studies [14, 15, 16, 17, 18], the participants performed only one task – language comprehension – and, although this complex task can arguably be broken down into simpler tasks, this question was not explicitly investigated by the authors. In contrast, Sudre et al. [19] explicitly tasked participants with answering yes/no questions about objects. Even though the original paradigm of Sudre et al. [19] results in task-dependent brain recordings, the authors average the brain recordings for the same stimulus across tasks and learn predictive models only based on the semantics of the objects. While averaging over repetitions of the same stimulus can boost the signal-to-noise ratio, it likely loses the task-dependent information in the brain recordings. Here we reanalyze the original task-dependent single-repetition data from Sudre et al. [19] to investigate the task-dependent brain recordings using predictive models that include representations of both the object and the question.

One previous work uses a predictive model to investigate task effects [11], and is thus closest to ours. In this work, the authors asked participants to attend to one of two object categories in natural scene stimuli. The authors then learn two separate models, each of which is trained to predict the fMRI recordings of participants in one of the 2 tasks as a function of the stimuli representations. They then compare the learned weights of the 2 models to conclude that each task-specific model puts more emphasis on those stimulus features that are related to the task. In contrast to this work, we integrate both the task and the stimulus representations into a single zero-shot learning framework, which allows us to predict brain recordings corresponding to novel tasks and stimuli. Additionally, we predict MEG recordings which have a 2000-times finer temporal resolution than the fMRI recordings used by Cukur et al. [11], which allows us to localize the task effect in time.

The work of Nastase et al. [12] also use a computational approach to investigate task effects. These authors account for the task directly in their computational model by constructing a different representational dissimilarity space [20] for each of two tasks, and then comparing these to the representational dissimilarity space of brain recordings. The representational spaces of the stimuli are entirely task-dependent and do not incorporate the stimulus semantics. This is a limitation because this model is not able to investigate the relationship between the brain recordings and a possible interaction between the task and stimulus. Moreover, similarly to Cukur et al. [11], Nastase et al. [12] predict fMRI recordings, limiting the investigation of the task effect in time.

## 3 Methodology

### 3.1 Brain data

We aim to study the effect of a task on the brain representation of a stimulus, when the stimulus is shown while performing the task. To this end, investigating the magnetoencephalograpy (MEG)

dataset presented in Sudre et al. [19], which contains 20 different question tasks, makes for an excellent case study and was provided upon our request.

In this experiment, subjects were asked to perform a question-answering task, while their brain activity was recorded using MEG. Figure 1 illustrates the experimental paradigm. Subjects were first presented with a question (e.g., *"Is it manmande?"*), followed by 60 concrete nouns, along with their line drawings, in a random order. Each stimulus was presented until the subject pressed a button to respond *"yes"* or *"no"* to the initial question. Once all 60 stimuli are presented, a new question is shown for a total of 20 questions. Thus we have a total of 60 stimuli $\times$ 20 questions $= 1200$ examples.

MEG samples the amplitude of the magnetic field induced by neuronal firing at 306 sensors positioned on the scalp of a subject every millisecond. The data were preprocessed using standard MEG preprocessing procedures (details in Appendix A). We analyze the data from the beginning of the stimulus word presentation (i.e. 0ms) to 800ms, to avoid contributions to the brain signal from the participant's button-press (median response time across stimuli is 913ms, averaged across subjects). We further downsample the recordings in time by averaging non-overlapping 25ms windows, resulting in data of size 306 sensors $\times$ 32 time windows . We analyze data from 6 of the original 9 subjects. Data from 3 subjects were excluded because of missing trials.

## 3.2 Selecting representations for questions and stimuli

To study the effect of the question on the meaning representation of a word, we first need a way to represent both the semantics of the question and the word. We created two types of word and question representations: one type derived from a pretrained bidirectional model of stacked transformers (BERT) [21], which is a popular model used for question-answering tasks, and a second type derived from Amazon Mechanical Turk (MTurk) of people answering questions about concrete nouns. We find that the MTurk representations significantly outperform the BERT ones in the prediction tasks outlined in the following sections, and so we focus on the MTurk representations in the main text and provide a detailed description of the BERT features and related results in Appendix D. One possible explanation for why representations from BERT perform worse is that BERT may lack commonsense knowledge related to perceptual and visual properties of objects that is necessary to answer the questions in our experiment (e.g. *"Is it bigger than a car?"*). In fact, prior work has shown that BERT representations are deficient of object attributes that are related to questions similar to ours [22] and of other physical commonsense knowledge [23].

The Mechanical Turk data was originally collected by Sudre et al. [19] and was provided at our request. Participants on MTurk were shown a set of 1000 words (e.g., *"bear", "house"*) and were requested to answer 218 questions about them (e.g., *"Is it fragile?", "Can it be washed?"*) on a scale from 1 to 5 ("definitely not" to "definitely yes"). In this dataset, 60 out of the 1000 presented words and 20 out the 218 questions corresponded to the stimuli and questions shown during the brain recording experiment. A complete list of words and questions is shown in Appendix H.

Using this dataset, we define the representation of a word as a vector containing the MTurk responses for that word to all 198 questions not in the experiment (see Figure 2). Moreover, we define the task (i.e. question) representation as a vector containing the MTurk responses for 60 words which are not in the experiment. Using more words did not result in improved performance on the validation set. We purposefully excluded the questions and words in the brain experiment from these representations. Note that [19] used the same word representations, but to the best of our knowledge, we are the first to represent question semantics as a collection of answers.

With permission from Sudre et al. [19], we provide the MTurk representations of the stimuli and questions in `https://github.com/otiliastr/brain_task_effect`. We further provide the MTurk human-judgments for all 1000 words, and the BERT representations discussed in Appendix D.

## 3.3 Hypotheses

Next, we formulate several hypotheses of how the question integrates with the stimulus in order to give rise to a task-dependent meaning representation. First, we will introduce the notation used to define the hypotheses, as well as the concrete models described in the next section. Using the notation in Table 1, we propose the following hypotheses, also shown in Figure 3:

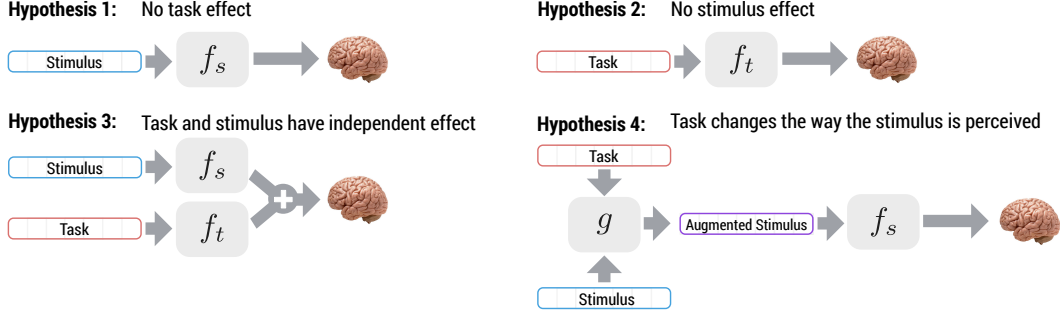

Figure 3: Proposed task-stimulus integration hypotheses.

**Hypothesis 1 (no task effect):** The brain activity is not affected by the task and can mostly be explained by the stimulus. Thus, we can approximate the elicited brain activity as: $b = f_s(s)$.

**Hypothesis 2 (no stimulus effect):** The brain activity is not affected by the stimulus and can mostly be explained by the task. Thus, we can approximate the elicited brain activity as: $b = f_t(s)$. While this hypothesis may not predict the brain activity the best, it will allow us to localize the task effect.

**Hypothesis 3 (additive):** Both the stimulus and the task affect the brain activity, but their contributions are independent: $b = f_s(s) + f_t(t)$.

**Hypothesis 4 (interactive):** The brain activity is well explained by the stimulus, but the task changes the way the stimulus is perceived: $b = f_s(t \otimes s)$. We can think of this as the task focusing *attention* on particular features of the stimulus that are relevant to the task (e.g., in answering the question *"Is it bigger than a car?"* for the stimulus *"dog"*, we pay more attention to the features of *"dog"* that are related to size, and ignore others such as color). This hypothesis aligns with the conclusions of Cukur et al. [11] that a task emphasizes those semantic dimensions of the stimulus that are relevant to the task. We use the notation $\otimes$ to represent generically any type of augmentation, and in Section 3.4 we describe in detail the forms of attention used in our experiments.

### 3.4 Predicting brain activity under different hypotheses

We next formulate models to represent the parametric functions $f_s$ and $f_t$ in the proposed hypotheses and to learn the parameters that best predict the brain activity. Our notation is summarized in Table 1. In the rest of this paper, we refer to the hypotheses using the abbreviations H1, H2, H3 and H4.

#### 3.4.1 Models

The functions $f_s$ and $f_t$ can be represented using any regression models that map from a feature space to the brain activity space. Prior work [14, 24, 25, 26] has shown that simple multivariate regression models such as *ridge regression* are reliable tools for predicting brain activity from stimulus features and are able to achieve good accuracy. For this reason, we will adopt the ridge regression setting for modeling $f_s$ and $f_t$. In ridge regression, we model the output of a function $f$ as a linear combination of the input features: $\hat{y} = f(x) = xW$, where $W$ is a parameter matrix. $W$ is trained to minimize the loss function $\|Y - XW\|_F^2 + \lambda\|W\|_F^2$, consisting of the mean squared error of the predictions and a regularization term on the parameters to avoid overfitting. Here $X$ and $Y$ represent the training inputs and targets, respectively, stacked together, $\|.\|_F$ denotes the Frobenius norm, and $\lambda > 0$ is a tunable hyperparameter representing the regularization weight. In our setting, the targets of the prediction $Y$ consist of the MEG recording of the brain activity, $Y_b$, described in Table 1. However, the inputs $X$ depend on the hypothesis being tested, as we describe further.

**Hypothesis 1:** Under a *no task effect* hypothesis, we predict the brain activity as a function of the stimulus features only, $Y_b = f_s(X_s) = X_s W_s$, where $W_s \in \mathbb{R}^{F_s \times LT}$. The objective function is:

$$\min_{W_s} \|Y_b - X_s W_s\|_F^2 + \lambda\|W_s\|_F^2 \tag{1}$$

**Hypothesis 2:** Under a *no stimulus effect* hypothesis, we predict the brain activity as a function of the task features only, $Y_b = f_t(X_t) = X_t W_t$, where $W_t \in \mathbb{R}^{F_t \times LT}$. Our objective function becomes:

$$\min_{W_t} \|Y_b - X_t W_t\|_F^2 + \lambda\|W_t\|_F^2 \tag{2}$$

Table 1: Notation used in defining the proposed hypotheses and models.

| | | | |
|---|---|---|---|
| $N_s$ | num. unique stimuli in experiment, 60 | $\hat{b}$ | predicted brain activity; $\hat{b} \in \mathbb{R}^{LT}$ |
| $N_t$ | num. unique tasks in experiment, 20 | $X_s$ | stimuli representations, stacked |
| $R$ | total num. repetitions, over all stimuli, 1200 | | for all repetitions; $X_s \in \mathbb{R}^{R \times F_s}$ |
| $L$ | space dimension of the brain activity, 306 | $X_t$ | task representations, stacked |
| $T$ | time dimension of the brain activity, 32 | | for all repetitions; $X_t \in \mathbb{R}^{R \times F_t}$ |
| $F_s$ | num. features in stimulus representation, | $Y_b$ | recorded brain activity, stacked |
| | 198 for MTurk; 768 for BERT | | for all repetitions; $Y_b \in \mathbb{R}^{R \times LT}$ |
| $F_t$ | num. features in task representation, | $f_s$ | function mapping from $s$ to $\hat{b}$; |
| | 60 for MTurk; 768 for BERT | | $f_s : \mathbb{R}^{F_s} \to \mathbb{R}^{LT}$ |
| $s$ | stimulus representation; $s \in \mathbb{R}^{F_s}$ | $f_t$ | function mapping from $t$ to $\hat{b}$; |
| $t$ | task representation; $t \in \mathbb{R}^{F_t}$ | | $f_t : \mathbb{R}^{F_t} \to \mathbb{R}^{LT}$ |

**Hypothesis 3:** Under an *additive effect* hypothesis, we predict the brain activity as the sum of the stimulus contribution and task contribution: $Y_b = f_s(X_s) + f_t(X_t) = X_s W_s + X_t W_t$. Note that this is equivalent to a single regression function $f(X_s, X_t) = [X_s, X_t] \cdot [W_s; W_t]$, where $[X_s, X_t] \in \mathbb{R}^{R \times (F_s + F_t)}$ is a concatenation of the stimulus and task features, and $W = [W_s; W_t] \in \mathbb{R}^{(F_s + F_t) \times LT}$ is a concatenation of their corresponding weight matrices. Thus, the objective can be written as:

$$\min_{W_s, W_t} \|Y_b - [X_s, X_t] \cdot [W_s; W_t]\|_F^2 + \lambda \|[W_s; W_t]\|_F^2 \qquad (3)$$

**Hypothesis 4:** Under an *interactive effect* hypothesis, we predict the brain activity as a function of the *augmented* stimulus features. The intuition is that the task *augments* the features that are relevant. In this work, we consider an implementation of the augmentation using *soft attention* [27], in which the task reweighs the contribution of the stimulus features. To simplify the notation in the following formulations, we will use $t$ and $s$ to refer to both the identity and the representation of a task and a stimulus in the experiment. Each task $t$ is associated with a set of attention parameters $a_t \in \mathbb{R}^{F_s}$ that rescale the original stimulus features when the stimulus $s$ is presented under question $t$. Thus, the augmented stimulus features under question $t$ become $\bar{s} = a_t \otimes s$, where $\otimes$ represents element-wise multiplication. The augmented stimuli for all training examples can be stacked together in an a matrix $X_{\bar{s}}$, and used as input to a ridge regression model, similar to H1:

$$\min_{W_s} \|Y_b - X_{\bar{s}} W_s\|_F^2 + \lambda \|W_s\|_F^2 \qquad (4)$$

The attention vectors $a_t$ can be precomputed or learned along the regression parameters, as follows:

**H4.1. Precomputed attention:** The MTurk features have interpretable dimensions for both tasks and stimuli, which enables us to directly compute the hypothesized relevance of different stimuli dimensions to each task. As described in Section 3.2, each semantic dimension of a word corresponds to one of the $F_s = 198$ non-experimental questions (see Figure 2). Given this relationship, we compute the attention parameters for every stimulus presented under task $t$ as $a_t = \texttt{softmax}([a_{t, \tilde{t}_j}])$ for $j \in \{1, \ldots, F_s\}$, where $\tilde{t} \in \mathbb{R}^{F_t}$ is a representation of a non-experimental question, and $a_{t, \tilde{t}} = \texttt{cosine\_similarity}(t, \tilde{t})$. We observe that this precomputed attention indeed emphasizes semantically-relevant word features. For example, the word features with highest attention for the question *"Is it made of metal?"* are *"Is it silver?"* and *"Is it mechanical?"*. The top 5 word features with highest attention for each question are provided in Appendix E.

**H4.2. Learned attention:** We learn the attention parameters together with the regression parameters with the objective of predicting the brain recordings as accurately as possible. A direct approach would be to learn a different set of attention parameters $a_t$ for every task $t$. However, since our goal is to be able to make *zero-shot* predictions for tasks and stimuli never seen during training, we instead learn how to map the features of the task to an attention vector. In our experiments we did so by learning an attention matrix $A \in \mathbb{R}^{F_t \times F_s}$, such that $a_t = \sigma(tA)$, where $\sigma(.)$ represents the sigmoid function, applied element-wise. Putting all pieces together, our objective function becomes:

$$\min_{W_s, A} \|Y_b - \sigma(X_t A) X_s W_s\|_F^2 + \lambda \|W_s\|_F^2 + \lambda_A \|A\|_F^2 \qquad (5)$$

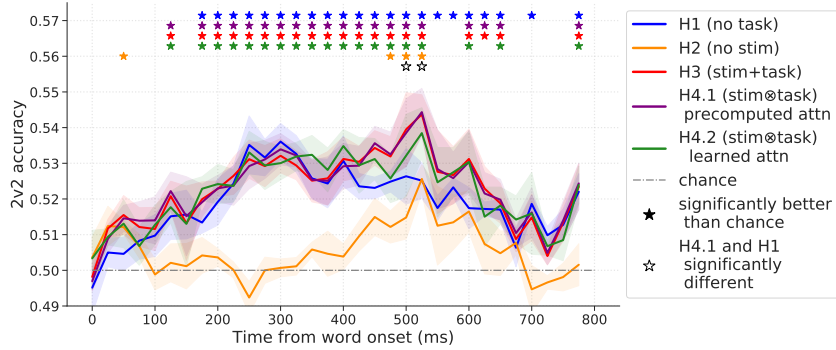

Figure 4: Performance of all hypotheses at predicting the MEG recordings in 25ms windows, averaged over sensors. We show the mean and std. error over subjects. The task effect is mostly localized to $475 - 550$ms. Hypotheses that incorporate both the stimulus and task perform similarly across time.

### 3.4.2 Training and evaluation

We next train and evaluate all models. Our goal is to predict brain recordings for any new task and word (i.e. zero-shot). Thus, we train all models using *leave-k-out* cross-validation, in which we leave out all training examples that correspond to task-stimuli pairs that contain either a task or a word that will be used for testing. We choose the regularization parameters via nested cross-validation.

We evaluate the predictions from each model by using them in a classification task on the held-out data, in the *leave-k-out* setting. The classification task is whether we are able to match the brain data predictions for two heldout task-stimuli pairs to their corresponding true brain data. This task has been previously proposed for settings with low signal-to-noise ratio [14]. The classification is repeated for each leave-k-out fold and an average classification accuracy is obtained for each sensor-timepoint. We refer to this accuracy as *2v2 accuracy*. The theoretical chance performance is $0.5$. A more detailed explanation about this metric can be found in Appendix B. Further details about the train/validation/test splitting, parameter optimization, hyperparameter tuning and preventing overfitting can be found in Appendix C.

Our code with all training and evaluation details is available at `https://github.com/otiliastr/brain_task_effect`.

## 4 Results and discussion

### 4.1 Effect of incorporating question task semantics

**Time window results.** We present the 2v2 accuracy per 25ms time window of all tested hypotheses in Figure 4. The time points for which each accuracy significantly differs from chance are indicated with a ⋆ symbol (one-sample t-test, $0.05$ significance level, FDR controlled for multiple comparisons [28]). We observe that the hypothesis that only considers the question task semantics (H2) performs significantly better than chance in one early time window ($50 - 75$ms) and much later during $475 - 550$ms. The remaining hypotheses also perform better than chance in the same $475 - 550$ms window, but we observe that during the majority of that time H3 and H4.1 perform significantly better than H1 (paired t-test, $0.05$ significance level, FDR controlled for multiple comparisons; significance for the H3-H1 comparison and all other pairwise comparisons are shown in Supplementary Figure 10 in Appendix F). We conclude that incorporating the question task semantics can improve the prediction of MEG recordings. Note that all discussed times are measured relative to stimulus onset.

**Sensor-timepoint results.** We investigate the task effect further by comparing the contribution of the question-specific precomputed attention and the word features to the accuracy of H4.1 by computing the 2v2 accuracy in two special cases: (1) when the two tested word-question pairs share the same word (i.e. $(q_1, w_1)$ vs $(q_2, w_1)$), higher-than-chance accuracy is attributed to the precomputed attention features; (2) when the two tested word-question pairs share the same question (i.e. $(q_1, w_1)$ vs $(q_1, w_2)$), higher-than-chance accuracy is attributed to the word features. These results are presented per sensor-timepoint in Figure 5, where only higher-than-chance accuracies across participants are shown

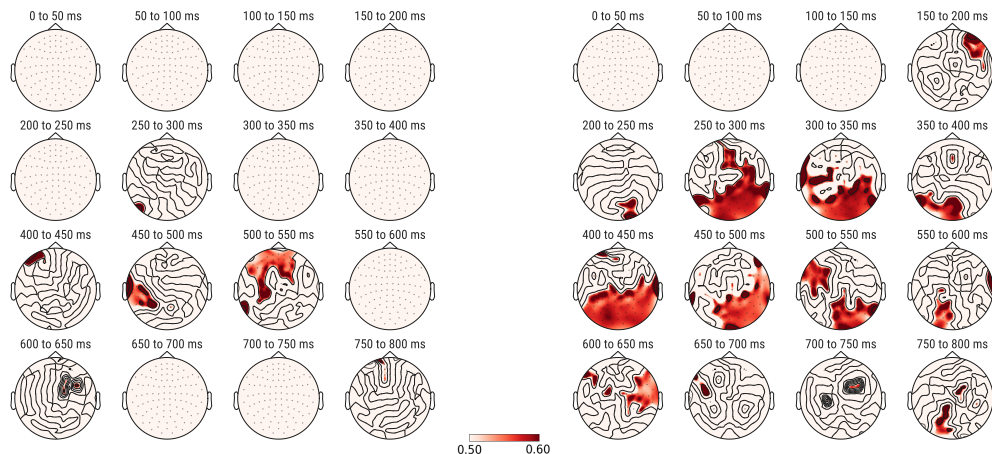

Figure 5: Mean 2v2 accuracy across subjects of predicting sensor-timepoints in 50ms windows using H4.1, when predicting the brain recordings for two word-question pairs that share the same word (Left) and the same question (Right). Displayed accuracies are significantly greater than $0.5$. The main question contribution appears in the frontal and temporal lobes during $400 - 550$ms, whereas the word contribution is distributed across the occipital and temporal lobes during $200 - 650$ms.

(one-sample t-test, 0.05 significance level, FDR controlled for multiple comparisons). The results are visualized using MNE-Python [29]. The main contribution of the question-specific attention appears between 400-550ms, localized to the frontal and the left temporal lobes. The contribution of the stimulus features is more distributed, both in time and space. The effect of word semantics begins at $150$ms and extends until the end of the considered time, with major contributions in the occipital lobes ($200 - 600$ms) and temporal lobes ($400 - 550$ms, $600 - 650$ms). For ease of visualization, here we present results for 50ms time windows. The results for 25ms time window align with the presented effects and are provided in Appendix F.

**Beyond MTurk representations.** We experimented with substituting the Mechanical Turk word and question representations with features extracted from BERT [21], which are used by many state-of-the-art methods across several natural language processing problems. In summary, we find that the BERT token-level word embeddings can be a good substitute of the MTurk embeddings, but the question representations of pretrained BERT do not appear to align as closely to the question semantics in the brain. These results can be found in Appendix D. We also tested replacing the stimulus representations with random vectors, which resulted in chance performance.

## 4.2    Comparison of task-stimulus interaction hypotheses

We further test which of the 3 hypothesized types of task-stimulus interaction (i.e., independent in H3, precomputed attention in H4.1, or learned attention in H4.2) best explains the observed MEG recordings. We observe that there is no significant difference among these hypotheses when averaging over the performance in all sensors (significance shown in Supplementary Figure 10).

**Sensor-timepoint results.** A group of sensors in the occipital lobes are significantly better predicted by H3 than by H4.1 at $200-250$ms (paired t-test, $0.05$ significance level, FDR controlled for multiple comparisons). This is when semantic processing of a word begins, so H3 may outperform H4.1 here because H3 has an independent contribution from the word representation. Both H3 and H4.1 perform significantly better than chance during $450 - 500$ms, and there are different sensor groups in the frontal lobe that are significantly better predicted by each hypothesis than the other. This suggests that this time point may contain both independent and interactive contributions of the task. We lastly observe that H4.1 outperforms H3 in the left temporal lobe during $600 - 650$ms. This localization suggests that the word and question semantics may interact in this time period, rather than be processed independently. These results are shown in Supplementary Figure 11.

**Learned attention.** There are no significant differences between H4.2 and H4.1, and the learned attention is highly similar to the precomputed one (Pearson correlation of $0.69$ between the pairwise cosine distances of the task-specific precomputed and learned attention parameters; more details in

Appendix E). This suggests that either the precomputed attention is one optimal way to combine the stimulus and task in predicting the recordings, or that more samples are needed to learn a better one. To further understand the effect of the sample size, we also evaluated H3 and H4.2 with varying amounts of training data, and found that both models perform increasingly better with more samples. The results and discussion are included in Appendix G.

### 4.3 Effect size

We note that the magnitudes of the presented effects (i.e. accuracies, differences between hypotheses) are limited due to the small amount of data and the underlying difficulty of analyzing single-trial MEG data. The accuracies we observe are on par with other reported single-trial MEG accuracies [16]. Other work has mitigated the low signal-to-noise ratio of single-trial MEG by averaging the recordings corresponding to different repetitions of the same stimulus [19] or grouping 20 examples together for a 20v20 classification task [16]. Neither is applicable here because our data does not contain repetitions of the same question-stimulus pair, and our zero-shot setting would require us to hold out a large portion of our training set if we were to evaluate on 20 stimulus-question pairs.

In the absence of these options, we have taken careful precautions to validate our results (by evaluating our models on held-out data in a cross-validated fashion) and evaluated the significance of the model performances and differences between them, and corrected for multiple comparisons. We trust that the effects we have shown to be significant are indeed true, but we note that there may be effects that we are not able to reveal due to limited power and hope that neuroscientists will apply our methods in the future to larger datasets with multiple repetitions.

### 4.4 Discussion and relation to previous results

Taken together, our results point to a robust effect of the question task semantics on the brain activity during $475 - 550$ms. We also find an effect of the interaction between the question and stimulus semantics during $600 - 650$ms, localized to the temporal lobe. The temporal lobes are implicated in semantic processing [1, 30, 31, 32] and specifically in maintaining relevant lexical semantic information for the purposes of integration [33]. Since this effect occurs past the time when a word is thought to be processed (i.e. up to 600ms), it may be related to maintaining specific semantic dimensions that help answer the question (the median response time across participants is 913ms). In addition to being localized to the temporal lobes, the earlier question effect is also found in the frontal lobes, which are thought to support attention [34, 35]. A task effect that is related to attention is consistent with findings from [11, 12]. Our results expand these previous findings by characterizing the temporal dynamics of the task-stimulus interactions.

## 5 Conclusions and future work

We propose a computational framework for comparing different hypotheses about how a task affects the meaning of an observed stimulus. The hypotheses are formulated as prediction problems, where a model is trained to predict the brain recordings of a participant as a function of the task and stimulus representations. We show that incorporating the semantics of a question into the predictive model significantly improves the prediction of MEG recordings of participants answering questions about concrete nouns. The timing of the effect coincides with the end of semantic processing for a word, as well as times when the participant is deciding how to answer the question.

These results suggest that only the end of semantic processing of a word is task-dependent. This finding may inspire new NLP training algorithms or architectures that keep some computation task-independent, in contrast to current transfer learning approaches for NLP that tune all parameters of a pretrained model when training to perform a specific task [21]. Moreover, future work can extend our methods to incorporate representations of tasks and stimuli from powerful neural networks that are augmented with improved commonsense knowledge [22], which would eliminate the need for human-judgment annotations. Furthermore, only one of the tested hypotheses (H4.1) is experiment-dependent, while all others can be applied to data from any neuroscience experiment, as long as task and stimulus feature representations can be obtained. Our results pose a challenge for future research to formulate new hypotheses for earlier effects on processing as a function of the task and stimuli.

## Broader impact

Our work pursues questions about the function of an average person's brain and makes contributions to basic science. We do not foresee a societal benefit or disadvantage to specific groups of people.

## Funding and competing interests

This material is based upon work supported by the DARPA D3M program, the NSF Graduate Research Fellowship, the Google Faculty Research Award, start-up funds in the Machine Learning Department at Carnegie Mellon University, and the AFOSR through research grants FA95501710218 and FA95502010118. The authors declare no competing interests.

## Acknowledgments

We thank Gustavo Sudre for collecting the MEG dataset and Dean Pomerleau for collecting the human-judgment Mechanical Turk dataset.

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
