[Supplementary Material]

# Appendix

## A  Data preprocessing

**MEG data preprocessing.**  The data were first preprocessed using the Signal Space Separation method (SSS) [37] in order to isolate the signal components that originate inside of the sensor array. This method was followed by its temporal extension (tSSS) to align the measurements of the head position before each block to a common space. The MEG signal was then filtered using a low-pass filter at 150Hz and notch filters at 60Hz and 120Hz to remove contributions from electrical line noise and other very high frequency noise. Next, the Signal Space Projection method was applied to remove eye blinks, residual movement, and other artifacts.

**Input and output normalization.**  When training our prediction models, both the model inputs and the targets are normalized as follows. For the input features (which could be word features, question features or both, depending on the hypothesis), we z-score each feature $x_i$ along the sample dimension by assigning $x_i \leftarrow \frac{x_i - \text{mean}(x_i)}{\text{std}(x_i)}$, such that each feature $x_i$ has mean 0 and standard deviation 1 across the samples. Similarly, we z-score every sensor-timepoint of the outputs across samples. However, to make sure our evaluation is correct and no information about the test data has been leaked during training, the mean and standard deviation used when z-scoring is calculated only over the training data.

## B  2v2 metric

There are several metrics that can be used to directly measure how close the brain activity prediction is from the ground truth, such as *Euclidean distance*, *cosine distance*, *percent of variance explained*. In our experiments, we use these metrics to choose the tunable model parameters (e.g., regularization $\lambda$). Another way to compare the hypotheses is by trying to match the predicted left-out brain responses to their corresponding ground truth, as introduced in Mitchell et al. [14], and illustrated in Figure 6. Having 2 left out repetitions, with predictions $\hat{b}_1$ and $\hat{b}_2$, and corresponding ground truth $b_1$ and $b_2$, we calculate two scores: $score_1 = dist(\hat{b}_1, b_1) + dist(\hat{b}_2, b_2)$ and $score_2 = dist(\hat{b}_1, b_2) + dist(\hat{b}_2, b_1)$, where the distance used in our experiments is cosine distance. If $score_1 < score_2$, we match $\hat{b}_1 \leftrightarrow b_1$ and $\hat{b}_2 \leftrightarrow b_2$ (correct match, accuracy 1), otherwise we match $\hat{b}_1 \leftrightarrow b_2$ and $\hat{b}_2 \leftrightarrow b_1$ (wrong match, accuracy 0). We further refer to this as the *2v2 accuracy*. Note that in our setting, we hold out more than 2 repetitions (i.e. word-question combinations), but in order to compute this metric, we take multiple combinations of 2 repetitions. Under this metric, chance performance is 50%.

## C  Training details

**Train/test splits.**  The next step is to train and evaluate the proposed models. For this, we need to separate our data into a train set and a test set. Since our dataset contains very few samples, as it is usually the case in neuroscience, we adopt the common *leave-k-out* cross-validation approach [14, 19, 26], in which the data is repeatedly split into 2 groups: one containing $k$ repetitions for test, and one with $R - k$ repetitions used for training. Each training set is further split into 2 subgroups using a similar approach: one for training, and one for parameter validation. A model is trained on the inner training set using multiple hyperparameters, and the ones with the best average validation accuracy are selected. Using the best hyperparameters, we retrain on the train+validation data, and compute the final accuracy on the test set. A common choice for $k$ in approaches that average the stimulus repetitions [14, 19] is 2, because this allows us to compare the brain activities for two left out stimuli (as described in the next paragraph), while training on as much data as possible. However, since we want our models to perform zero-shot learning and to be able to make predictions for both words and questions that have not been seen during training, we leave out from training 2 stimuli with all their 20 repetitions under different questions, but also 2 questions with all 60 words about which this question was asked (i.e. a total of $2 \times 20 + 60 \times 2 = 160$ examples). Out of these 160 examples, we only test the model performance on the word-question pairs for which neither the word nor the question appear in training. We do this type of splitting both when performing train/test splitting, and for train/validation splitting.

Figure 6: 2v2 metric. The predictions for two repetitions, $\hat{b}_1$ and $\hat{b}_2$, are being matched to their corresponding true brain activities, $b_1$ and $b_2$. The match is performed based on the distances between each of the predictions, to each of the true brain activities: $score_1 = dist(\hat{b}_1, b_1) + dist(\hat{b}_2, b_2)$ and $score_2 = dist(\hat{b}_1, b_2) + dist(\hat{b}_2, b_1)$. If $score_1 < score_2$, we match $\hat{b}_1 \leftrightarrow b_1$ and $\hat{b}_2 \leftrightarrow b_2$ (correct match, obtaining an accuracy of 1.0), otherwise we match $\hat{b}_1 \leftrightarrow b_2$ and $\hat{b}_2 \leftrightarrow b_1$ (wrong match, accuracy 0.0). The distance used in our experiments is cosine distance.

**Optimization.** While H1, H2 and H3 can be solved in closed form, we use the Cholesky decomposition approach provided in the Python `scikit-learn` package [38] for computational reasons. In H4, we need to optimize the parameters of the functions $g$ and $f_s$ together, and thus we implemented this using the TensorFlow framework [39] and trained end-to-end using the Adam optimizer [40] with default parameters and learning rate 0.001.

**Parameters and hyperparameters.** The only hyperparameters in our framework are the regularization parameters $\lambda$ (for all hypotheses) and $\lambda_A$ (only for H4.1). Their values were chosen from the set of values $\{10^{-5}, 10^{-4}, ..., 10^7\}$ using the train/validation/test splitting described above. We also allowed the model to select different $\lambda$ per sensor-timepoint, but found that using the same value for outputs is more stable and leads to better validation accuracy overall. Moreover, we found conducting the parameter validation using the cross-validation setting described above on all subjects and all hypotheses to be prohibitive, and thus we performed the hyperparameter tuning for each hypothesis on a single subject, which was then excluded from testing. Regarding the number parameters, each hypothesis has a different number of parameters, depending on the size of the of the inputs and extra attention parameters, with H3 > H4.1 > H1 = H4.2 > H2. To ameliorate any effects of overfitting, we allow each hypothesis to choose its own regularization parameters.

## D   BERT features and corresponding results

**Model details**   BERT is a bidirectional model of stacked transformers that is trained to predict whether a given sentence follows the current sentence, in addition to predicting a number of input words that have been masked [21]. We use the base pretrained model provided by the Hugging Face Transformers library [41]. This model has 12 layers, 12 attention heads, and 768 hidden units.

**Extracting BERT word and question features.**   We first apply WordPiece tokenization to each of our 60 word stimuli and questions. To extract the word features, we pass the tokens corresponding to each word stimuli into the BERT model and extract the corresponding token-level embeddings. We use these token-level embeddings as the BERT word representations in the following experiments. If any word contains more than 1 token, it is assigned the average of the corresponding token-level embeddings.

Figure 7: Pairwise cosine distances among the question representations from MTurk and the two types of BERT question representations. The MTurk representations appear to better cluster semantically-similar questions together.

Figure 8: Comparisons of 2v2 accuracies of predictions computed using MTurk vs. BERT features in each hypothesis. Mean accuracy and standard error across subjects plotted. Points where the means across subjects are significantly different are marked with a ★ symbol.

To extract the question features, we pass the tokenization of each question separately into BERT, with a '[CLS]' token and a '[SEP]' token appended to respectively the beginning and end of the tokenized list. This is common practice with inputting multi-word sequences into BERT. We then extracted the hidden layer activations from the CLS token at the last hidden layer, as well as the pooled output. In the pretrained model, the pooled representation of a sequence is a transformed version of the embedding of the [CLS] token, which is passed through a hidden layer and then a tanh function.

**Question feature dissimilarity.** There are many different choices for where to extract the question representations from BERT – e.g. the CLS token representation from any of the 12 hidden layers, the pooled output, or a mean or max pooling across all token representations in any of the layers. To settle on which BERT representation may be best suited to predict the MEG recordings, we first visualize how similar the representations for our 20 questions are under different BERT representations (see Figure 7). We observe that the MTurk representations appear to better cluster semantically-similar questions together. The representations from the CLS token at the last layer appear to cluster together sentences that share words, whereas the pooled output representations appear to lead to at least two larger clusters that correspond to questions related to animacy and size. We therefore conduct experiments using the BERT pooled output embeddings as the question representations.

**Performance against MTurk features.** Features extracted from BERT for both the stimuli and questions perform significantly worse than the Mechanical Turk (MTurk) features in several timewindows across hypotheses (see Figure 8) (paired t-test, significance level 0.05, FDR controlled for multiple comparisons [29]). Some of the difference in performance may be due to the large difference in dimensionality between the BERT and MTurk features. The BERT features have much higher dimensionality than the MTurk features, which may lead to more overfitting when using the BERT features. However, this is likely not the only cause for the difference, as the question representations from BERT appear much worse at predicting the MEG recordings in the $450 - 550$ms timewindow, where we show the question/task semantics contribute most to the MEG recordings. It is likely that the pretrained BERT model is not able to compose the input words in a way that is as brain-aligned as the question representations from Mechanical Turk. It would be an interesting future direction to fine-tune BERT on a question-answering task and compare the performance of the pretrained question representations with that of the fine-tuned BERT representations.

# E  Attention

**Relevant word features determined by the precomputed attention.** The precomputed attention is described in Section 3.4.1, and assigns a different relevance score (between 0 to 1) to each word feature for each question. Here we list the top 5 most relevant word features (i.e. with highest attention scores) for each of the 20 experimental question, as determined by the precomputed attention. The features are listed in decreasing order of importance (the first is the most important).

1. 'Can you hold it?':
   - CAN IT BE EASILY MOVED?
   - IS IT LIGHTWEIGHT?
   - WOULD YOU FIND IT IN A HOUSE?
   - CAN YOU TOUCH IT?
   - CAN YOU BUY IT?

2. 'Can you hold it in one hand?':
   - CAN IT BE EASILY MOVED?
   - IS IT LIGHTWEIGHT?
   - WOULD YOU FIND IT IN A HOUSE?
   - DO YOU HOLD IT TO USE IT?
   - CAN YOU BUY IT?

3. 'Can you pick it up?':
   - CAN IT BE EASILY MOVED?
   - IS IT LIGHTWEIGHT?
   - CAN YOU BUY IT?
   - WOULD YOU FIND IT IN A HOUSE?
   - DO YOU HOLD IT TO USE IT?

4. 'Is it bigger than a loaf of bread?':
   - IS IT HEAVY?
   - IS IT TALLER THAN A PERSON?
   - IS IT LONG?
   - DOES IT COME IN DIFFERENT SIZES?
   - IS IT USUALLY OUTSIDE?

5. 'Is it bigger than a microwave oven?':
   - IS IT TALLER THAN A PERSON?
   - IS IT HEAVY?
   - IS IT BIGGER THAN A BED?
   - IS IT LONG?
   - IS IT USUALLY OUTSIDE?

6. 'Is it bigger than a car?':
   - IS IT BIGGER THAN A BED?
   - IS IT BIGGER THAN A HOUSE?
   - IS IT TALLER THAN A PERSON?
   - IS IT HEAVY?
   - IS IT LONG?

7. 'Can it keep you dry?':
   - DOES IT PROVIDE SHADE?
   - IS IT A BUILDING?
   - DOES IT PROVIDE PROTECTION?
   - CAN YOU TOUCH IT?
   - ARE THERE MANY VARIETIES OF IT?

8. 'Could you fit inside it?':
    - IS IT BIGGER THAN A BED?
    - IS IT TALLER THAN A PERSON?
    - DOES IT PROVIDE SHADE?
    - IS IT A BUILDING?
    - IS IT BIGGER THAN A HOUSE?

9. 'Does it have at least one hole?':
    - DOES IT HAVE A FRONT AND A BACK?
    - IS IT SYMMETRICAL?
    - DOES IT HAVE PARTS?
    - DOES IT COME IN DIFFERENT SIZES?
    - DOES IT HAVE INTERNAL STRUCTURE?

10. 'Is it hollow?':
    - IS IT A BUILDING?
    - DOES IT HAVE FLAT / STRAIGHT SIDES?
    - DOES IT OPEN?
    - DOES IT COME IN DIFFERENT SIZES?
    - CAN YOU TOUCH IT?

11. 'Is part of it made of glass?':
    - DOES IT HAVE WIRES OR A CORD?
    - DOES IT USE ELECTRICITY?
    - IS IT A BUILDING?
    - DOES IT HAVE FLAT / STRAIGHT SIDES?
    - DOES IT HAVE WRITING ON IT?

12. 'Is it made of metal?':
    - IS IT SILVER?
    - IS IT MECHANICAL?
    - WAS IT INVENTED?
    - IS IT SHINY?
    - DOES IT HAVE A HARD OUTER SHELL?

13. 'Is it manufactured?':
    - WAS IT INVENTED?
    - DOES IT HAVE WRITING ON IT?
    - DOES IT HAVE FLAT / STRAIGHT SIDES?
    - CAN YOU USE IT?
    - CAN YOU BUY IT?

14. 'Is it manmade?':
    - WAS IT INVENTED?
    - DOES IT HAVE FLAT / STRAIGHT SIDES?
    - DOES IT HAVE WRITING ON IT?
    - CAN YOU USE IT?
    - ARE THERE MANY VARIETIES OF IT?

15. 'Is it alive?':
    - IS IT CONSCIOUS?
    - IS IT AN ANIMAL?
    - IS IT WARM BLOODED?
    - DOES IT HAVE EARS?
    - IS IT WILD?

16. 'Was it ever alive?':
    - IS IT AN ANIMAL?
    - IS IT WILD?
    - CAN IT BITE OR STING?
    - IS IT CURVED?
    - IS IT CONSCIOUS?

17. 'Does it grow?':
    - IS IT WILD?
    - IS IT CONSCIOUS?
    - IS IT AN ANIMAL?
    - CAN IT BITE OR STING?
    - IS IT WARM BLOODED?

18. 'Does it have feelings?':
    - IS IT CONSCIOUS?
    - DOES IT HAVE EARS?
    - DOES IT HAVE A BACKBONE?
    - IS IT WARM BLOODED?
    - DOES IT HAVE A FACE?

19. 'Does it live in groups?':
    - IS IT AN ANIMAL?
    - CAN IT JUMP?
    - IS IT A HERBIVORE?
    - IS IT WILD?
    - IS IT CONSCIOUS?

20. 'Is it hard to catch?:
    - IS IT FAST?
    - IS IT A PREDATOR?
    - IS IT AN ANIMAL?
    - CAN IT JUMP?
    - IS IT USUALLY OUTSIDE?

**Learned attention vs. precomputed attention**   We want to compare the learned attention parameters in H4.2 to the precomputed ones in H4.1. To this end, we compute the pairwise cosine distances across the precomputed question-wise attention, across the learned question-wise attention. The resulting cosine distance matrices are visualized in Figure 9. To quantify the similarity between the two, we compute the Pearson correlation between the upper-triangles of the two matrices, which comes out to $0.69$. This indicates that the learned attention, that is randomly initialized, learns to combine the question and word features in a way that is very similar to the precomputed attention.

# F  Supplementary results

**Pairwise comparisons across all hypotheses.**   We present the pairwise comparisons of 2v2 accuracy performance across all tested hypotheses in Figure 10. All timepoints where there is significant difference between the performances of the two displayed hypotheses are marked with a star (paired t-test, significance level 0.05, FDR controlled for multiple comparisons). The hypothesis that does not incorporate information about the stimulus (H2) performs significantly worse than all hypotheses that do during $250 - 400$ms. The hypothesis that does not incorporate information about the task (H1) performs significantly worse than two hypotheses that do (H3 and H4.1) in time windows between $450$ and $550$ms. There is no significant difference in the performances across timewindows of all hypotheses that are a function of both the stimulus and task (H3, H4.1, and H4.2)

Figure 9: Pairwise cosine distances across the question-wise attention in H4.1 (Left) and H4.2 (Right). The question-wise attention in H4.2 is an average over participants. The Pearson correlation between these matrices (upper-triangle only) is 0.69, indicating a high degree of correspondence between the precomputed attention (Left) and the mean learned attention (Right).

Figure 10: Pairwise comparisons of 2v2 accuracy performance across all tested hypotheses. All timepoints where there is significant difference between the performances of the two displayed hypotheses are marked with a star.

Figure 11: Significant differences in performance between H4.1 and H3 per sensor-timepoint (H4.1 accuracy - H3 accuracy). The red points are those where H4.1 significantly outperforms H3, and the blue are those where H3 significantly outperforms H4.1. We're only displaying the significant differences for those timepoints where H4.1 performs significantly better than chance (Left) and where H3 performs significantly better than chance (Right).

**Sensor-timepoint comparison of H4.1 against H3.** We present the significant differences in performance between H4.1 and H3 per sensor-timepoint in Figure 11. We have plotted H4.1 accuracy - H3 accuracy (the red points are those where H4.1 significantly outperforms H3, and the blue are those where H3 significantly outperforms H4.1). We're only displaying the significant differences for those timepoints where H4.1 performs significantly better than chance (Left) and where H3 performs significantly better than chance (Right). The discussion of these results can be found in the main text in Section 4.2.

**Sensor-timepoint results for H4.1 for** 25**ms windows.** We present the sensor-timepoint results for H3 for 25ms time-windows in Figure 12. They follow the general trend of the results from 50ms time-windows presented in Figure 5.

Figure 12: Mean 2v2 accuracy across subjects of predicting sensor-timepoints in 25ms timewindows using H4.1, when predicting the brain recordings for two stimulus-question pairs that share the same word (Top) and the same question (Bottom). The displayed accuracies are significantly greater than chance.

# G   Experiments varying the training sample size

We also performed some experiments to compare our different hypotheses using varying amounts of training data. Since it would be difficult to collect more data beyond the 20 questions and 60 words, we performed this experiment by reducing the amount of data we allow the model to train on.

Figure 13: Experiments with various amounts of training data.

We trained H4.2 with increasing amounts of data and tested its performance. We also tested H3, which is a simpler model that we expect to learn with fewer samples. The results are shown in Figure 13. H3 continues to improve as we add more examples, up to the maximum (i.e. 1044 samples = 58words×18 questions). This suggests that even this simpler model may benefit from more training data. H4.2 also appears to improve with more samples, however it is less clear whether the performance peak has been reached or whether this is due to the difficulty of the optimization problem. These results are even more clear in the time interval $450 - 600$ms, where we expect the two models to perform the best according to the results in Figure 4.

# H Stimuli and questions

**Questions in experiment:**

1. Can you hold it?
2. Can you hold it in one hand?
3. Can you pick it up?
4. Is it bigger than a loaf of bread?
5. Is it bigger than a microwave oven?
6. Is it bigger than a car?
7. Can it keep you dry?
8. Could you fit inside it?
9. Does it have at least one hole?
10. Is it hollow?
11. Is part of it made of glass?
12. Is it made of metal?
13. Is it manufactured?
14. Is it manmade?
15. Is it alive?
16. Was it ever alive?
17. Does it grow?
18. Does it have feelings?
19. Does it live in groups?
20. Is it hard to catch?

**Stimuli in experiment:** dog, horse, arm, eye, foot, hand, leg, apartment, barn, church, house, igloo, arch, chimney, closet, door, window, coat, dress, pants, shirt, skirt, bed, chair, desk, dresser, table, ant, bee, beetle, butterfly, fly, bottle, cup, glass, knife, spoon, bell, key, refrigerator, telephone, watch, chisel, hammer, pliers, saw, screwdriver, carrot, celery, corn, lettuce, tomato, airplane, bicycle, car, train, truck

**All questions asked on Mechanical Turk:**

- Is it an animal?
- Is it a body part?
- Is it a building?
- Is it a building part?
- Is it clothing?
- Is it furniture?
- Is it an insect?
- Is it a kitchen item?
- Is it manmade?
- Is it a tool?
- Can you eat it?
- Is it a vehicle?
- Is it a person?
- Is it a vegetable / plant?
- Is it a fruit?
- Is it made of metal?
- Is it made of plastic?
- Is part of it made of glass?
- Is it made of wood?
- Is it shiny?
- Can you see through it?
- Is it colorful?
- Does it change color?
- Is one more than one colored?
- Is it always the same color(s)?
- Is it white?
- Is it red?

- Is it orange?
- Is it flesh-colored?
- Is it yellow?
- Is it green?
- Is it blue?
- Is it silver?
- Is it brown?
- Is it black?
- Is it curved?
- Is it straight?
- Is it flat?
- Does it have a front and a back?
- Does it have a flat / straight top?
- Does it have flat / straight sides?
- Is taller than it is wide/long?
- Is it long?
- Is it pointed / sharp?
- Is it tapered?
- Is it round?
- Does it have corners?
- Is it symmetrical?
- Is it hairy?
- Is it fuzzy?
- Is it clear?
- Is it smooth?
- Is it soft?
- Is it heavy?
- Is it lightweight?
- Is it dense?
- Is it slippery?
- Can it change shape?
- Can it bend?
- Can it stretch?
- Can it break?
- Is it fragile?
- Does it have parts?
- Does it have moving parts?
- Does it come in pairs?
- Does it come in a bunch/pack?
- Does it live in groups?
- Is it part of something larger?
- Does it contain something else?
- Does it have internal structure?
- Does it open?

- Is it hollow?
- Does it have a hard inside?
- Does it have a hard outer shell?
- Does it have at least one hole?
- Is it alive?
- Was it ever alive?
- Is it a specific gender?
- Is it manufactured?
- Was it invented?
- Was it around 100 years ago?
- Are there many varieties of it?
- Does it come in different sizes?
- Does it grow?
- Is it smaller than a golfball?
- Is it bigger than a loaf of bread?
- Is it bigger than a microwave oven?
- Is it bigger than a bed?
- Is it bigger than a car?
- Is it bigger than a house?
- Is it taller than a person?
- Does it have a tail?
- Does it have legs?
- Does it have four legs?
- Does it have feet?
- Does it have paws?
- Does it have claws?
- Does it have horns / thorns / spikes?
- Does it have hooves?
- Does it have a face?
- Does it have a backbone?
- Does it have wings?
- Does it have ears?
- Does it have roots?
- Does it have seeds?
- Does it have leaves?
- Does it come from a plant?
- Does it have feathers?
- Does it have some sort of nose?
- Does it have a hard nose/beak?
- Does it contain liquid?
- Does it have wires or a cord?
- Does it have writing on it?

- Does it have wheels?
- Does it make a sound?
- Does it make a nice sound?
- Does it make sound continuously when active?
- Is its job to make sounds?
- Does it roll?
- Can it run?
- Is it fast?
- Can it fly?
- Can it jump?
- Can it float?
- Can it swim?
- Can it dig?
- Can it climb trees?
- Can it cause you pain?
- Can it bite or sting?
- Does it stand on two legs?
- Is it wild?
- Is it a herbivore?
- Is it a predator?
- Is it warm blooded?
- Is it a mammal?
- Is it nocturnal?
- Does it lay eggs?
- Is it conscious?
- Does it have feelings?
- Is it smart?
- Is it mechanical?
- Is it electronic?
- Does it use electricity?
- Can it keep you dry?
- Does it provide protection?
- Does it provide shade?
- Does it cast a shadow?
- Do you see it daily?
- Is it helpful?
- Do you interact with it?
- Can you touch it?
- Would you avoid touching it?
- Can you hold it?
- Can you hold it in one hand?
- Do you hold it to use it?
- Can you play it?
- Can you play with it?
- Can you pet it?
- Can you use it?

- Do you use it daily?
- Can you use it up?
- Do you use it when cooking?
- Is it used to carry things?
- Can you pick it up?
- Can you control it?
- Can you sit on it?
- Can you ride on/in it?
- Is it used for transportation?
- Could you fit inside it?
- Is it used in sports?
- Do you wear it?
- Can it be washed?
- Is it cold?
- Is it cool?
- Is it warm?
- Is it hot?
- Is it unhealthy?
- Is it hard to catch?
- Can you peel it?
- Can you walk on it?
- Can you switch it on and off?
- Can it be easily moved?
- Do you drink from it?
- Does it go in your mouth?
- Is it tasty?
- Is it used during meals?
- Does it have a strong smell?
- Does it smell good?
- Does it smell bad?
- Is it usually inside?
- Is it usually outside?
- Would you find it on a farm?
- Would you find it in a school?
- Would you find it in a zoo?
- Would you find it in an office?
- Would you find it in a restaurant?
- Would you find in the bathroom?
- Would you find it in a house?
- Would you find it near a road?
- Would you find it in a dump/landfill?
- Would you find it in the forest?
- Would you find it in a garden?
- Would you find it in the sky?
- Do you find it in space?
- Does it live above ground?
- Does it get wet?
- Does it live in water?
- Can it live out of water?
- Do you take care of it?
- Does it make you happy?
- Do you love it?
- Would you miss it if it were gone?
- Is it scary?
- Is it dangerous?
- Is it friendly?
- Is it rare?
- Can you buy it?
- Is it valuable?