[Reviews · NeurIPS 2020]

Review 1

Summary and Contributions: This paper investigates using machine learning and MEG data how the brain processes words conditionally on the task. The paper has clear merit although the effects revealed are small and technical novelty are moderate. Sharing the code is particularly appreciated.

Strengths: The paper is very well written, with adequate literature review and reasonable experimental results.

Weaknesses: Major concerns: - It is unfortunate that source localization results are not provided. Topographical maps in fig 5 are not easy interpret. - Attentional model is neat and is potentially interesting for the NeurIPS audience but unfortunately it does not lead to any improved restults. Doing a learning curve analysis should help to answer the question if "more samples are needed to learn a better one". - Using ADAM in this setting (only a few samples) seems dangerous. Using L-BFGS using gradients from tensorflow could potentially stabilize and accelerate the estimation. Minor concerns: - please number the equations - what software was used to analyse the MEG data? Brainstorm (Tadel et al.), Fieldtrip (Oostenveld et al.), MNE (Gramfort et al.)? - When you introduce a matrix or vector notation such as W provide the dimensions - in equation of H4.2 paragraph you minimize over W_s and A (A is missing in the \min) - use \|.\| rather than ||.|| for the norm in Latex - Using B and Y for the target is unnecessarily confusing, also having features call Q and S. I would stick to X_Q or X_S so features are always called X and target always Y. - use ^\top rather than ^T for transpose especially as you have T as a notation for time.

Correctness: ok

Clarity: well written

Relation to Prior Work: good

Reproducibility: Yes

Additional Feedback:


Review 2

Summary and Contributions: This paper presents a re-analysis of the MEG experiment of Sudre et al (2012), where participants were tasked with responding to a question about the meaning of an object concept word (e.g. “Can you hold it?” CARROT (respond yes/no)). In the original Sudre et al analysis, the focus was on testing the predictive power of different perceptual and semantic feature models of the concept word for the MEG data. In the current study, the focus is on the role of the task question that precedes the concept word, and in particular whether and how the semantics of the task question modulates the subsequent processing and neural activity time-locked to the stimulus word. This is an interesting neurocognitive question, as it sheds light on how lexical-semantic representation and access can be modulated by the preceding context, and how the timing of processing of the target concept word that is independent of the task demands relates to the timing of the processing that involves integrating that conceptual knowledge with the task requirements in order to respond on the task. To analyze the data, the authors construct vector-based semantic models of both the concept words and the task questions, using human responses from separate questions and concepts where the participants rated the truth of the task questions for the concepts. This yields vector-based representations that reflect the semantic content of the questions and the concepts, where semantically similar questions are represented with similar vectors and semantically similar concepts are represented as similar vectors. The authors use these stimulus feature models to predict the MEG brain data. In particular, they construct different predictive linear models, corresponding to different hypotheses about how task question and concept word processing affect the neurocognitive signal. They consider a model that uses the task question representations alone, a model that uses the concept word representations alone, and a model that concatenates the task and concept representations in the predictive model (such that the brain activity is predicted from some linear combination of the task and concept features). In addition, they consider two “interactive” models, where task representations modulate concept representations through a soft attention mechanism. The first of these uses precomputed attention weights, whereas the second are learned in conjunction with the other linear model weights when fitting the predictive model to the brain data. The results show that the concept-word-only predictive model yields the most parsimonious prediction of the brain data up to what is presumed to be the end of word lexical processing, at around 475 ms, with the task-only model being significant in the 475-525 ms timeframe. In addition, the models combining task and concept features show their best performance relative to the concept-word-only model in this later time frame, and in particular one of the attention based models significantly outperforms the concept-word-only model at 500-525 ms. This pattern of results suggests a particular time-course of processing in this experimental paradigm, with lexico-semantic access of the word, independent of task requirements, preceding the integration of the task requirements with the word representation in order to perform the task. UPDATE: I have read an considered the rebuttal document and the other reviews. I am not a fan of the "grouping 20 examples together for a 20v20 classification tasK" approach to evaluation that the authors mention, which has always seemed to me a kludge to artificially increase reported accuracies by trying to ignore between-item variability (not unlike the "language as a fixed effect fallacy" in psycholinguistics). The authors have not responded to my comment on the unnatural experimental design (understandable given the space limitations). However, I still have an overall favourable opinion of this work.

Strengths: This study presents a competent set of analyses on the Sudre et al dataset, and I found all aspects of the methodology (design of the stimulus models with the MTurk data, construction of the four types of predictive models corresponding to different neurocognitive hypotheses, and the use of an attention parameter matrix to model interactive effects of task question on concept representation) convincing. The results of the analysis make sense – the initial transient effect of the task-alone model can be interpreted as residual processing of the cue question, and the results nicely support the claim that processing of the concept word initially happens independently of subsequent task effects.

Weaknesses: The experimental paradigm used seems a bit unnatural. Rarely are we asked a question about an unspecified object’s properties (“Can you hold it”) and *then* get given the object name. This lack of ecological validity may mean that the results do not generalize well to lexico-semantic processes in question answering more generally (and the paper does imply in a number of places that the task is about question answering). Although I liked the paper and also would like to see more work featuring neuroimaging data at NeurIPS, I wonder if it is not more suited to a cognitive neuroscience venue. The use of soft attention in combining feature models in predicting the MEG data is interesting, but apart from this the focus of the paper is perhaps not particularly methodologically novel. It is a pity that the analyses with the BERT model do not seem to work out effectively, as this might be of interest to understanding how BERT represents questions about object concepts. The final section of the paper makes reasonable suggestions about how BERT might be incorporated into the analysis more effectively, but that is not included in the current work. Although the results make sense, are significant, and are interesting, the reported differences in the H1 and H4.1 models in the later time period are not huge (a difference of about 1%-1.5% in prediction accuracy) and overall the decoding performance is low (~53%). This, combined with the small number of experimental participants (6) makes me wonder about the robustness of the reported findings.

Correctness: Yes, I think the analyses are correct and appear to have been competently implemented.

Clarity: Yes. I have no concerns about the quality of the writing, and the paper is easy to understand. Typo: line 181: “thes regularization weight” - > “the regularization weight”

Relation to Prior Work: In general, the prior work is well-covered, and in particular the paper does a good job situating itself with respect to other work using similar linear decoding analysis frameworks (e.g. the work of the Gallant and Mitchell labs and others). Given the scientific focus of the paper, there could perhaps be better coverage of the cognitive neuroscience literature on lexical-semantic representation and retrieval, such as work by Beth Jeffries, Jeff Binder, Sharon L. Thompson-Schill inter alia., and a better situation of the results of the paper with respect to neurocognitive theories of word processing.

Reproducibility: Yes

Additional Feedback:


Review 3

Summary and Contributions: The authors propose a predictive model to examine the relationship between brain recordings and stimulus properties and explicitly investigated task effects at the same time. It is the first computational model to predict brain recordings as a function both of the task and the semantics of the observed stimulus. Such research is beneficial for future studies on question-answering in the neuroscience area.

Strengths: - "Modeling Task Effects on Meaning Representation in the Brain" is an important study. It is good to know the relationship between them which might give some insights for researchers on model designing. - The authors use MEG recordings which 2000-times finer temporal resolution than the fMRI recordings and thus allow to localize the task effect in time. - The hypotheses are thoroughly explored in the context of exploring relationship of task and stimulus.

Weaknesses: - Apart from ridge regression, there are some regression models like lasso regression the authors might try. - For Table 1, the notation should be aligned with its description. - Overall, I am quite curious why 2v2 accuracies for all hypotheses are just slightly better than the random chance. Why is that? Correct me if I am wrong, thanks.

Correctness: To best of my knowledge, yes.

Clarity: Yes.

Relation to Prior Work: Yes.

Reproducibility: Yes

Additional Feedback: Overall, there are few new things for the methodology part. But what the authors study is quite important and interesting. Therefore, initial rating can be changed as further information given. %%%%%Post-rebuttal After reading the rebuttal and comments from other reviewers, I will increase my score to 6. It is an interesting and important problem to understand task effects both for ML and neuroscience community. My only concern is the number of specimens used in the experiments. To acquire a more general picture of task effects, more samples may be needed. Lastly, thanks for answering all my questions.

[Author Response · NeurIPS 2020]

We thank the reviewers for the thoughtful comments and attempt to address their questions, space permitting.

**Methodological contributions [all reviewers].** We would like to clarify our *methodological* contributions and their
significance as follows: **(1)** we provide the first methodology that can predict brain recordings as a function of *both* the
observed stimulus and question task. This is important because it will not only encourage neuroscientists to formulate
mechanistic computational hypotheses about the effect of a question on the processing of a stimulus, but also enable
neuroscientists to test these different hypotheses against each other by evaluating how well they can align with brain
recordings. While we have implemented and compared several hypotheses for this effect, and have found some to be
better than others, parts of the MEG recordings remain to be explained by future hypotheses. We hope neuroscientists
will build on our method to formulate and test such future hypotheses. We will make our code publicly available to
facilitate this. **(2)** we perform all learning in a zero-shot setting, in which neither the stimulus nor the question used
to evaluate the learned models is seen during training (i.e. not just as the specific stimulus-question pair but also in
combination with any other question/stimulus). Note that this is not the case in previous work that examines task effects,
and we are the first to demonstrate how zero-shot learning can be applied successfully to this question. This is important
for scientific discovery because it can test the generalization of the results beyond the experimental stimuli and tasks.

**Effect size [all reviewers].** We acknowledge that the magnitudes of the presented effects (i.e. accuracies, differences
between hypotheses) are small, due to a limited amount of data and the underlying difficulty of analyzing single-trial
MEG data. The accuracies we observe are on par with other reported single-trial MEG accuracies[36]. Other work
has mitigated the low signal-to-noise ratio of single-trial MEG by averaging the recordings corresponding to different
repetitions of the same stimulus[30], or grouping 20 examples together for a 20v20 classification task[36]. Neither is an
option for us because our data does not contain multiple repetitions of the same question-stimulus pair, and our zero-shot
setting would require us to hold out a large portion of our training set if we were to evaluate on 20 stimulus-question
pairs. In the absence of these options, we have taken careful precautions to validate our results (by evaluating our
models on held-out data in a cross-validated fashion) and evaluated the significance of the model performances and
differences between them, and corrected for multiple comparisons. We trust that the effects we have shown to be
significant are indeed true, but we agree that there may be effects that we are not able to reveal due to limited power and
hope that neuroscientists will apply our methods in the future to larger datasets with multiple repetitions.

**Optimizing H4.2 [R1].** Following R1's suggestion, we trained H4.2 with increasing
amounts of data. We also tested H3, which is a simpler model that we expect to learn
with fewer samples. To the right, we show that H3 continues to improve as we add
more examples, up to the maximum (i.e. $1044$ samples $= 58$words$\times 18$ questions). This
suggests that even this simpler model may benefit from more training data. H4.2 also
appears to improve with more samples, however it is less clear whether the performance
peak has been reached or whether this is due to the difficulty of the optimization problem.
Here we present results for times where we expect the two models to perform the best
$(450 - 600\text{ms}, \text{Fig.4})$, and will include the full figures in the paper. For the H4.2
optimizer, we chose Adam because it is a common choice for non-linear problems, such
as H4.2. We agree that further investigation into different optimizers, such as L-BFGS,
may further improve the H4.2 results. We thank R1 for the suggestion and will test other
optimizers and include the results in the appendix.

**Connection to word processing literature [R2].** While we have focused the related work on studies that are method-
ologically similar to ours or investigate task effects, we agree that relating the results to word processing theories will
strengthen the paper and thank R2 for the suggestion. We will incorporate this in the discussion section.

**NeurIPS fit [R2].** We believe our contributions are of interest to both neuroscience and ML researchers, and NeurIPS
is the foremost venue where these communities come together. We will include a discussion of the following directions
that we hope our work inspires: **(1)** *new NLP architectures or training algorithms*: it is common to train an NLP
model to perform well at a specific task by tuning all model parameters, including the word embeddings. A more
brain-aligned method, as inspired by our findings that the question task affects the late stages of processing, would keep
some computation task-independent. Such a model may exhibit less catastrophic forgetting when learning new tasks.
**(2)** *better evaluation methods for AI models*: our finding that a question representation based on human judgements
outperforms BERT may interest the ML community that works on evaluating and incorporating spatial reasoning and
other common sense abilities into deep neural networks. **(3)** *new models for task-stimulus interaction in the brain*: the
growing computational neuroscience community at NeurIPS is particularly suited for designing such models.

**Other types of regularization [R3].** Previous work has found that as long as regularization parameters are properly
selected via cross-validation, as is done in our case, different regularization techniques lead to similar results for brain
prediction (Wehbe et al. 2015, Ann.Appl.Stat.). We also found that, when the Lasso regularization parameters were
carefully tuned, Lasso led to similar results to Ridge regression. However, the Lasso results were much more sensitive
to tuning and the optimization was slower. We will incorporate this justification for using Ridge regression in the paper.

**Additional comments [R1].** We used custom-built functions that are provided in the supplementary to analyze the
MEG data and MNE to visualize Fig. 5 (which we will properly cite). We agree that source localization would improve
the understanding of the task effect location, and we leave this to future neuroscientific research.

[Meta-Review · NeurIPS 2020]

Understanding how the tasks that we perform while perceiving a stimulus modulate brain activity is of wide interest to neuroscience. Reviewers found the experimental setup interesting, with the clear hypotheses about how tasks can impact neural activity. The small effect sizes observed were identified as a key limitation in drawing conclusions from this experiment. Reviewers found this worrisome particularly when coupled with marginal accuracies and few subjects. Reviewers identified that BERT results were not very conclusive, and significantly more could be done. This would make the paper much more interesting to an ML audience. Reviewers did not find the methods themselves to be particularly novel, but did find that the application and clear test was valuable. In general, the work is suggestive of how one might more deeply connect multitask models to neural data, something that has not occurred to date and could provide fertile ground for both creating new models and furthering our understanding of the brain.